# Simulated climate change, but not predation risk, accelerates *Aedes aegypti* emergence in a microcosm experiment in western Amazonia

Ana C. Piovezan-Borges[1]*, Francisco Valente-Neto[1], Wanderli P. Tadei[2], Neusa Hamada[2], Fabio O. Roque[1,3]

**1** Programa de Pós-Graduação em Ecologia e Conservação, Instituto de Biociências (INBIO), Universidade Federal de Mato Grosso do Sul (UFMS), Campo Grande, Mato Grosso do Sul, Brazil, **2** Coordenação de Biodiversidade, Divisão de curso em Entomologia, Instituto Nacional de Pesquisas da Amazônia (INPA), Manaus, Amazonas, Brazil, **3** Centre for Tropical Environmental and Sustainability Science (TESS), College of Science and Engineering, James Cook University, Cairns, Australia

\* anacpborges3@gmail.com

**Data Availability Statement:** All relevant data are within the manuscript and its Supporting Information files.

## Abstract

Climate change affects individual life-history characteristics and species interactions, including predator-prey interactions. While effects of warming on *Aedes aegypti* adults are well known, clarity the interactive effects of climate change (temperature and $CO_2$ concentration) and predation risk on the larval stage remains unexplored. In this study, we performed a microcosm experiment simulating temperature and $CO_2$ changes in Manaus, Amazonas, Brazil, for the year 2100. Simulated climate change scenarios (SCCS) were in accordance with the Fourth Assessment Report of Intergovernmental Panel on Climate Change (IPCC). Used SCCS were: Control (real-time current conditions in Manaus: average temperature is ~25.76°C ± 0.71°C and ~477.26 ± 9.38 parts per million by volume (ppmv) $CO_2$); Light: increase of ~1,7°C and ~218 ppmv $CO_2$; Intermediate: increase of ~2.4°C and ~446 ppmv $CO_2$; and Extreme: increase of ~4.5°C and ~861 ppmv CO2, all increases were relative to a Control SCCS. Light, Intermediate and Extreme SCCS reproduced, respectively, the B1, A1B, and A2 climatic scenarios predicted by IPCC (2007). We analyzed *Aedes aegypti* larval survivorship and adult emergence pattern with a factorial design combining predation risk (control and predator presence–*Toxorhynchites haemorrhoidalis* larvae) and SCCS. Neither SCCS nor predation risk affected *Aedes aegypti* larval survivorship, but adult emergence pattern was affected by SCCS. Accordingly, our results did not indicate interactive effects of SCCS and predation risk on larval survivorship and emergence pattern of *Aedes aegypti* reared in SCCS in western Amazonia. *Aedes aegypti* is resistant to SCCS conditions tested, mainly due to high larval survivorship, even under Extreme SCCS, and warmer scenarios increase adult *Aedes aegypti* emergence. Considering that *Aedes aegypti* is a health problem in western Amazonia, an implication of our findings is that the use of predation cues as biocontrol strategies will not provide a viable means of controlling the accelerated adult emergence expected under the IPCC climatic scenarios.

**Funding:** ACPB received part of fellowship from Coordenação de Aperfeiçoamento de Pessoal de Nível Superior (CAPES) and part from Fundação de Apoio ao Desenvolvimento do Ensino, Ciência e Tecnologia do Estado de Mato Grosso do Sul (Fundect). ACPB was supported by Fundo Brasileiro para Biodiversidade (FUNBIO) and Instituto Humanize grant. FVN received a Post-doctoral fellowship #88882.317337/2019-01 from CAPES; NH (307849/2014-7; 308970/ 2019-5) and FOR are Conselho Nacional de Desenvolvimento Científico e Tecnológico (CNPQ) research fellow. This study was also, partially, financed by CAPES - Finance Code 001. The views expressed in the present article are those of the authors and not necessarily those of any funding agencies. The funders had no role in study design, data collection and analysis, decision to publish, or preparation of the manuscript.

**Competing interests:** The authors have declared that no competing interests exist.

## Introduction

Climate change is among the main environmental concerns of this century [1]. Its effects on arthropod vectors have been stimulating intense research, due to the risks that such vectors may pose to human health. Mosquitoes are one of the main vectors of human diseases, globally causing more than 17% of all infectious diseases [2]. Worldwide, the geographic range of mosquitoes is expanding [3], and the number of vector-borne diseases has also increased in recent years [4]. The main mosquito-borne diseases, such as dengue and malaria cause some 700,000 deaths annually, and the large numbers of people infected often overloads health systems [2]. *Aedes aegypti* (*Ae. aegypti*) mosquitoes are one of the main disease vectors, being responsible for the transmission of dengue, yellow fever, Zika, and chikungunya viruses. Around the world, some 390 million people are infected with dengue virus each year [5]. The effects of climate change on adult disease vectors are well known [6, 7], because the virus is transmitted during this stage. However, few studies focus on larval life-history, despite it being well known that changes that occur in the environment of the larval stage, such as climate change, may shape the development and behavior of adults (this being known as a carry-over effect) [8].

Climate change affects biodiversity at multiple levels. It may cause shifts on biomass [9], metabolism and behavior [10], at the individual level and, at population level, it can alter species distribution via changes in local conditions. Consequently, community composition can be altered by climate change [11], changing ecosystems and food webs [12–14]. There are two ways in which predator-prey interactions are influenced by climate change. First, it can increase the metabolic rates of individuals, as a consequence of higher temperatures [12], affecting the ability of predators to forage, capture and handle prey. In this way, climate change may modify prey density (density-mediated interactions) [15]; Second, besides direct predation, climate change alter predator-prey interactions via production, transmission, and reception of chemical cues. Under such circumstances, both predator and prey may suffer reduction in their abilities to detect each other [16]. In predation risk situations, releases of chemical cues is common, and the detection of predator by prey through them can modify feeding behavior and/or development rates (trait-mediated interactions) [15].

Temperature increases can influence the metabolism, behavior and life-history traits of adult mosquitoes [17, 18], including speeding up development [19–21]. This may result in enhanced offspring production and, consequently, increase the number of people infected by etiologic agents transmitted by mosquitoes. For example, Ryan et al. [3] showed that warming increases the transmission risk of diseases caused by *Ae. aegypti* and *Aedes albopictus*. Meanwhile, predation risk can cause different responses in mosquitoes; adult female *Culex pipiens* increase dispersal distance in the presence of predators [22], while predation risk does not alter *Ae. aegypti* survivorship [23], even though it decreases adults lifespan [24]. In natural systems, individuals, populations and community dynamics are all affected by both abiotic and biotic factors [25]. Thus, it is essential to better understand the unexplored interactive effects of simulated climate change scenarios (SCCS) and predation risk on *Ae. aegypti* larval stage. Understanding the efficacy of predation risk in the development of a disease vector species under different SCCS can provide information on carry-over effects, and a perspective into the efficacy of using predation cues as biocontrol strategies.

Our overall goal was to understand the single and interactive outcomes of SCCS and predation risk on larval survivorship and adult emergence pattern of *Ae. aegypti*. It is important to consider this interaction in an environment that favors the development of this species, such as western Amazonia, where the climate is hot and humid throughout the year. Accordingly, we conducted an experiment in a microcosm simulating real-time climatic condition in Manaus (Control) and gradual increase in temperature and $CO_2$ in other three SCCS for this

city in the year 2100. We used as a predator *Toxorhynchites haemorrhoidalis* (*T. haemorrhoidalis* Diptera: Culicidae) larvae to investigate the effect of predation risk on *Ae. aegypti*. This microcosm simulates four climate change scenarios predicted by the Fourth Assessment Report (AR4) of Intergovernmental Panel on Climate Change (IPCC) [26].

It is widely known that temperature increases, within thermal tolerance, affects development and behavior of *Ae. aegypti* [27]. Predation risk alone would not affect directly prey survivorship, but might lead to changes in prey development and behavior, as well as phenotypic alterations [15, 28]. Although, effects of predation risk under climate change are uncertain, they can either accelerate, decrease or cause no change in prey behavior and life-history characteristics [16]. Accordingly, we hypothesized that the single or interaction effects of both ecological factors, SCCS and predation risk, would not affect *Ae. aegypti* larval survivorship, mainly since SCCS lie within the thermal tolerance of *Ae. aegypti* [27]. Our second hypothesis was that increase in climatic variables (temperature and $CO_2$) under SCCS would accelerate adult emergence of *Ae. aegypti*, and interactive effects of SCCS and predation risk would lead to earlier emergence. In this context, we discuss implications of our findings concerning the impact of predation risk on *Ae. aegypti* larvae reared under different SCCS to western Amazonia.

## Methods

### Simulated Climate Change Scenarios (SCCS)

The SCCS (microcosm) comprised of four chambers (4.05m x 2.94m), designed in accordance with the AR4-IPCC [26] recommendations to simulate temperature and $CO_2$ concentrations for the year 2100 in Manaus. The microcosm is located in the Center for Studies of Adaptations of Aquatic Biota of the Amazon (long-term project ADAPTA), installed in the Laboratory of Ecophysiology and Molecular Evolution at the National Institute for Amazon Research (LEEM/INPA), Manaus, Amazonas, Brazil.

The SCCS include: i) Control: real-time current conditions in Manaus, Amazonas, Brazil, the average temperature is 25.76 ± 0.71°C and the $CO_2$ concentration is 477.26 ± 9.38 parts per million by volume (ppmv). The other three scenarios reproducing respectively B1, A1B and A2 climatic conditions predicted by AR4-IPCC (2007) are: ii) Light: increase of ~1.7°C and ~218 ppmv $CO_2$, iii) Intermediate: increase of ~2.4°C and ~446 ppmv $CO_2$, and iv) Extreme: increase of ~4.5°C and ~861 ppmv $CO_2$. Temperature and $CO_2$ concentration of the Control SCCS varied instantaneously according to external values to capture real-time daily variation in Manaus. Values of temperature and $CO_2$ concentration for the Control SCCS were used to estimate values for Light, Intermediate and Extreme SCCS (S1 and S2 Figs). The SCCS was monitored automatically every 2 min to maintain temperature and $CO_2$ concentration values. Photoperiod was 12h light:12h dark, and humidity was approximately 80% in all chambers.

### Predator and prey

To study the effect of predation risk on *Ae. aegypti* larval survivorship and adult emergence pattern, we designed two treatments: control (without predation) and predation risk using *T. haemorrhoidalis* larvae as the predator. This species frequently coexists in natural and artificial environments with *Ae. aegypti* [29]. *Aedes aegypti* larvae use different strategies to avoid predators, such as seeking shelter in macrophytes roots [30]. In such habitats, they are under predation risk effects, which can lead prey to decrease the search for food resources and allocate energy in defense instead of development [31, 32]. Larvae of *T. haemorrhoidalis* were collected in Manaus and, prior to the experiment, were housed individually in cups with water and fed

daily with *Ae. aegypti* larvae until reaching the 3<sup>rd</sup> instar. Based on a pilot study, we estimated that each predator consumes one *Ae. aegypti* larva per day, and set this as the number to feed to the predators during the experiment. Predators were acclimatized in each SCCS for two days prior to the beginning of the experiment. We also acclimatized an additional two individuals for replacement purposes (in case a predator died).

Eggs of *Ae. aegypti* were obtained from colonies held by the Malaria and Dengue Laboratory at INPA. These colonies were established with wild-caught eggs collected in Manaus, using oviposition traps. *Aedes aegypti* eggs were collected with authorization and approval of the Brazilian Biodiversity Authorization and Information System (SISBIO; Permit 61563). We install multiple traps to obtain eggs in different private properties. Each owner gave us permission to install the traps. Fieldwork did not involve endangered or protected species. We placed filter paper to collect eggs of 4<sup>th</sup> and 5<sup>th</sup> generation adults from these colonies, and used these in the experiment. In each SCCS, the filter paper containing the eggs were placed in plastic containers until hatching occurred (± 17 hours). Following hatching, 60 first instar larvae were placed in each replication (see below).

## Experimental design

We designed a factorial experiment to test the effect of predation risk and SCCS on *Ae. aegypti* larval survivorship and adult emergence pattern (Fig 1). Survivorship was defined as larvae that survived until the adult stage, to calculate this we used the number of emerged adults divided by the initial number of larvae in each replicate. Adult emergence pattern was estimated via counting the number of adults emerged in each replicate on a daily basis, time to adult emergence was considered from hatching to adult emergence. We included four replicates for each combination of factors (predation risk and SCCS). The experimental units were plastic containers (20x30x6 cm) with distilled water and fish food TetraMin™ to provide *Ae. aegypti* feed.

At the beginning of the experiment, we filled the plastic containers with 600 mL of distilled water and placed the predator cage in the center of each container. The predator cage was a circular plastic container, 10.1 cm in diameter, sealed with nylon mesh to ensure water circulation, but still prevent entrance of *Ae. aegypti* into the predator enclosure (see Fig 1). The predator cage was placed in all experimental replicates (predation risk or control) to avoid any effect caused by the presence of the cage itself.

Then, to each replicate, we added one predator and 60 randomly selected *Ae. aegypti* first instar larvae. Each replicate received 0.0264g of food every two days. The density 0.1 larvae/mL and the amount of food were calculated to avoid effects related to intraspecific competition [33]. If evaporation occurred, water was added to the plastic containers to maintain the original water level. The 3<sup>rd</sup> instar *T. haemorrhoidalis* larvae were fed daily with one 4<sup>th</sup> instar *Ae. aegypti* larva, other than those used in the experiment. Predation of *Ae. aegypti* larva by *T. haemorrhoidalis* larva releases chemical cues that could be perceived by the other larvae in the container [23].

*Aedes aegypti* were maintained in the replicates until adult emergence, we recorded the daily adult emergence and total survivorship in each replicate.

## Statistical analysis

To evaluate whether the SCCS (Control, Light, Intermediate and Extreme), predation risk (predator and control), and their interaction, affected *Ae. aegypti* larval survivorship and adult emergence pattern, we performed a two-way ANOVA, while a *post hoc* least square means test was carried out if any of the tested factors tested were significant.

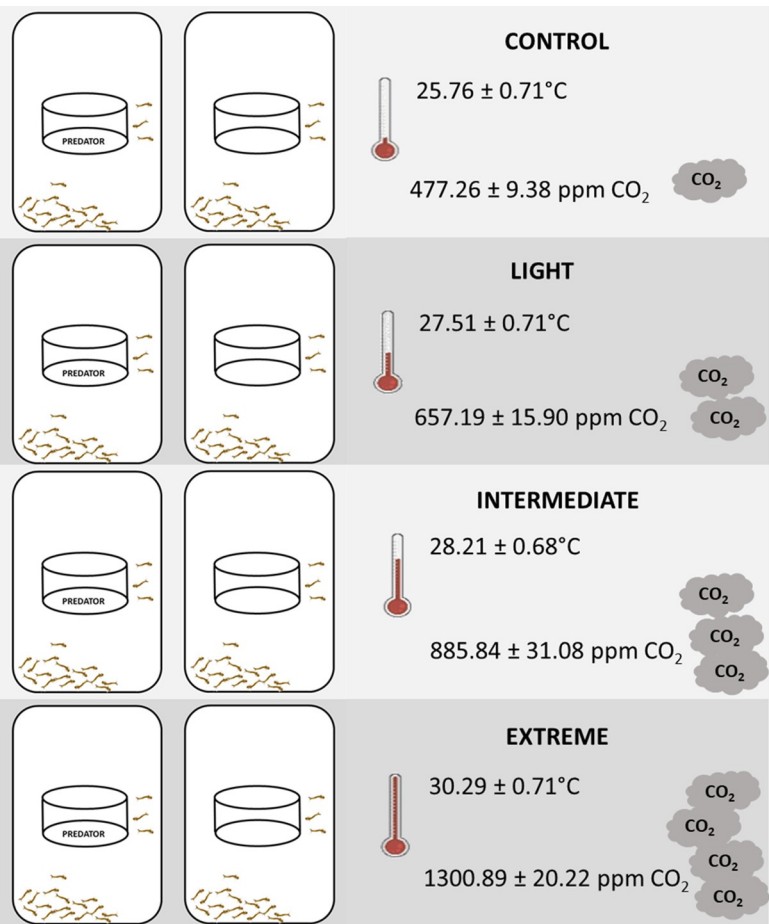

**Fig 1. Experimental design.** Experiment with two predation risk levels (predation risk–first column; and control–second column) in the four SCCS (Control, Light, Intermediate and Extreme—rows). Four replicates were used in the experiment. For each SCCS, the mean temperature and $CO_2$ concentration are shown. Humidity was 83.91 ± 2.10% in all scenarios.

Adult emergence pattern was evaluated using two complementary approaches. We used linear mixed model considering the replicates as a random factor, an autocorrelation function to control repetition across days, and SCCS and predation risk as fixed factors. This model is equivalent to a two-way ANOVA with repeated measures, and was used to detect if SCCS, predation and their interaction affected the response variable, i.e. number of emergences, controlling the repeated measures and time autocorrelation. Second, to better understand the effect of these predictors on emergence patterns without including time in our models as fixed factor (keeping enough degrees of freedom), we evaluated the distribution of emergence time through the estimation of skewness and kurtosis of the adult emergence per day for each replicate. Skewness measures horizontal asymmetry in data distribution relative to normal curve. A symmetrical distribution is indicated by a skewness coefficient of zero; positive and negative skewness values indicate the data are right-skewed and have a longer right tail, and left-skewed with a longer left tail, respectively. For example, a right-skewed would indicate an early emergence of *Ae. aegypti*, whereas a left-skewed would indicate a late emergence. Kurtosis measures vertical asymmetry in data distribution relative to a normal curve. Zero values indicate a normal curve (mesokurtic); positive values (leptokurtic) indicate that the shape of the curve is more peaked than the normal distribution; negative values (platykurtic) indicate that the shape

of the curve is flatter than the normal curve. A positive kurtosis would indicate that most larvae emerge at the same time, whereas a negative kurtosis would indicate a more evenly emergence distribution. Then, we applied a two-way ANOVA and a *post hoc* least square means test using skewness and kurtosis as response variables and SCCS and predation risk as predictors. Finally, we plotted emergence patterns per day. All analyses were carried out in the R environment [34].

## Results

### Survivorship

Mean *Ae. aegypti* larval survivorship was higher than 78% for all SCCS. Larval survivorship was not affected by SCCS (F = 2.77, $P$ = 0.0638), predation risk (F = 1.14, $P$ = 0.2957), or by interaction between them (F = 1.45, $P$ = 0.2517) (Fig 2).

### Emergence patterns

The first *Ae. aegypti* adult emergence was on day seven in the Extreme SCCS (n = 3 individuals), on day eight in the Intermediate SCCS (n = 11 individuals), while in the Light and Control SCCS the first adult emerged on day nine (n = 39 individuals; n = 3 individuals, respectively), with higher emergence rate in the Light SCCS. Adult *Ae. aegypti* emergence was not influenced by fixed factors SCCS ($F$ = 0.0536; $P$ = 0.9836), predation risk ($F$ = 0.0221; $P$ = 0.8819), or their interaction ($F$ = 0.0282; $P$ = 0.9936). However, adult emergence pattern measured by skewness and kurtosis revealed interesting results. SCCS significantly affected both skewness ($F$ = 4.287; $P$ = 0.0147) and kurtosis ($F$ = 4.905; $P$ = 0.0085). Emergence pattern in the Intermediate SCCS was not significantly different from the Control SCCS (df = 24; $P$ = 0.0584), but was significantly higher (right skewness) than the Light (df = 24; $P$ = 0.0269) and Extreme (df = 24; $P$ = 0.0291) SCCS (Fig 3). This indicates that most individuals emerged earlier in the Intermediate SCCS. Furthermore, emergence pattern estimated by kurtosis in the Intermediate SCCS was different from the other three SCCS (Control: df = 24, $P$ = 0.0449; Light: df = 24, $P$ = 0.0286; and Extreme: df = 24, $P$ = 0.0107) (Fig 4). The Intermediate SCCS was the only one with positive kurtosis (peak of frequency distribution), indicating that most individuals emerged on day ten, while in the other SCCS the adult emergence was more evenly distributed across several days (Fig 5).

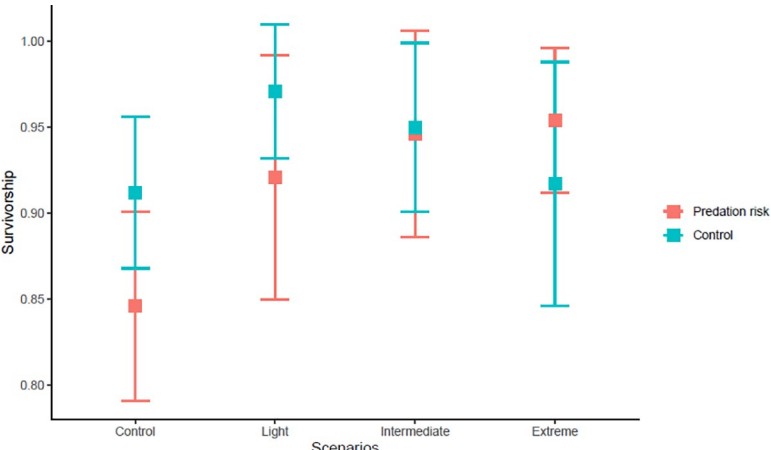

**Fig 2. *Aedes aegypti* larval survivorship.** Mean (confidence interval of 95%) of total larval survivorship of *Aedes aegypti* larvae after 14 days in four SCCS (Control; Light- increase of ~1.7˚C; Intermediate- increase of ~2.4˚C; and Extreme- increase of ~4.5˚C) in the presence (predation risk) and absence (control) of *Toxorhynchites haemorrhoidalis* predatory larva.

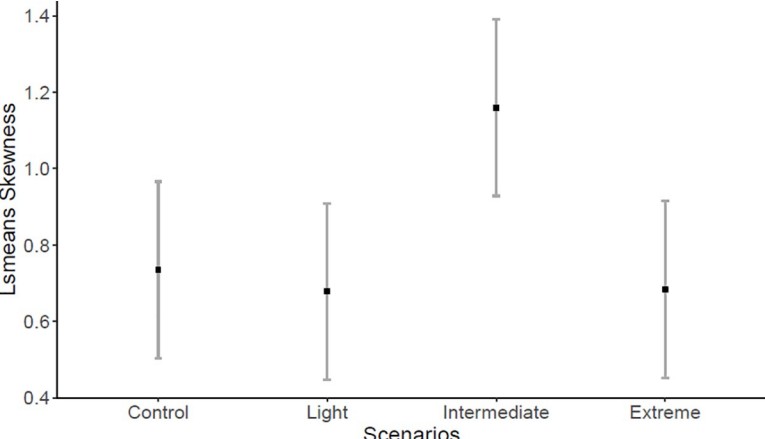

**Fig 3. Least squares mean (confidence interval of 95%) for skewness.** Emergence pattern estimated by skewness of *Aedes aegypti* between four SCCS (Control; Light- increase of ~1.7˚C; Intermediate- increase of ~2.4˚C; and Extreme-increase of ~4.5˚C).

## Discussion

We showed that simulated climate change scenarios accelerate development time of *Ae. aegypti* larvae, which agrees with previous studies in western Amazonia [21], with a emergence peak on a single day in Intermediate SCCS. We also found that *Ae. aegypti* larval survivorship was not affected by SCCS and predation risk, with larval survivorship rates being greater than 78% in all replicates, indicating the resilience of this species. We observed only SCCS did affect adult emergence pattern of *Ae. aegypti*, indicating that, in this study, climatic variables effects (temperature and $CO_2$ concentration) are stronger ecological driver than predation risk, particularly those related to chemical and visual cues.

Other studies have also reported that predation risk did not affect development time or survivorship *Ae. aegypti* larvae [23, 35], although, in some cases, an effect was perceived in the adult stage. For example, blood feeding success was higher in *Ae. aegypti* females exposed to predator risk during the larval stage [23]. Similarly, Chandrasegaran et al. [35] found

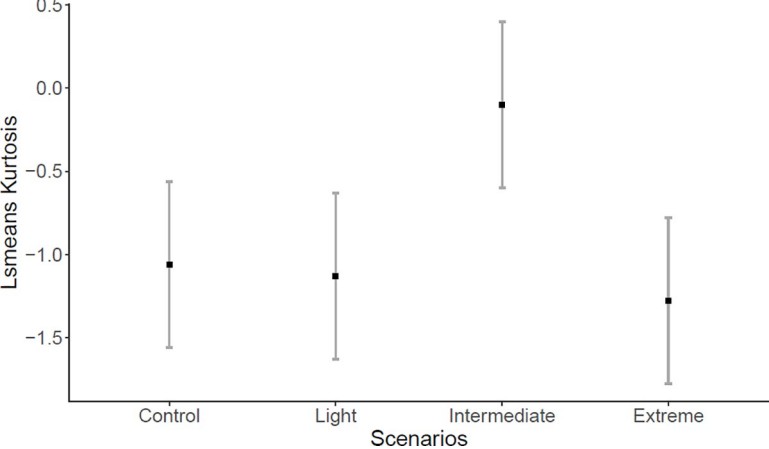

**Fig 4. Least squares mean (confidence interval of 95%) for kurtosis.** Emergence pattern estimated by kurtosis of *Aedes aegypti* between four SCCS (Control; Light- increase of ~1.7˚C; Intermediate- increase of ~2.4˚C; and Extreme-increase of ~4.5˚C).

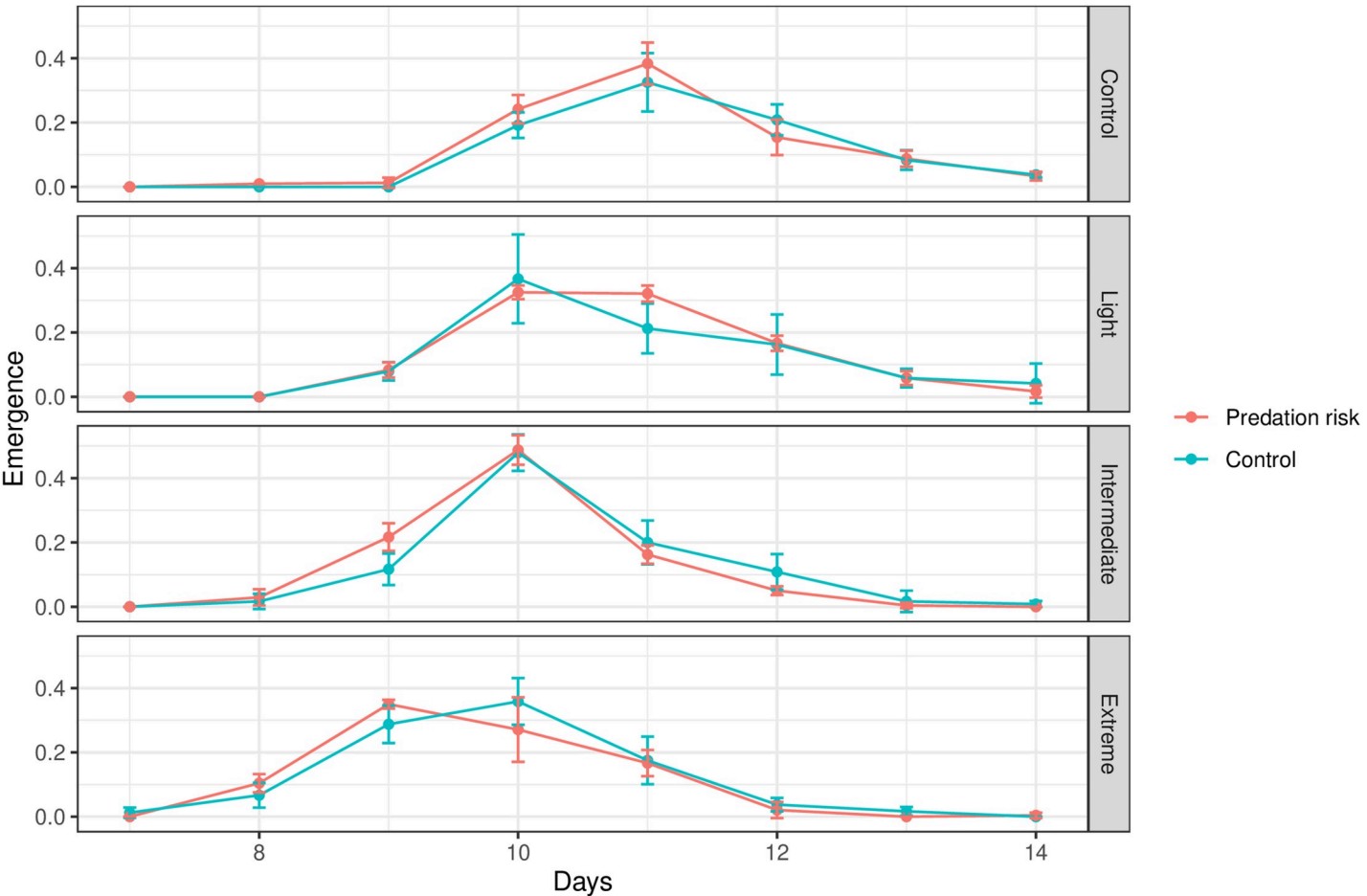

**Fig 5. Mean (±SE) adult emergence of *Aedes aegypti*.** Mean adult emergence in four SCCS (Control; Light- increase of ~1.7°C; Intermediate- increase of ~2.4°C; and Extreme- increase of ~4.5°C), and in two predation treatments: control and under predation risk (predator: *Toxorhynchites haemorrhoidalis* larva), from the first (day 7) until the last day (day 14) when all individuals reached adulthood.

interactive effects of predation risk, competition and food availability on teneral reserves in *Ae. aegypti* males. Also, interaction between predation risk and high intraspecific competition reduced the egg production of *Ae. aegypti* females [23]. In our study, we did not find any effect of predation risk on *Ae. aegypti* larval survivorship or adult emergence pattern, even in interaction with SCCS. These findings have implications for human health, as the impact of predation risk may not keep pace with the accelerated development of *Ae. aegypti* larvae under SCCS. Consequently, a warmer world will have more mosquitoes and an increase in vector-borne diseases [36].

*Toxorhynchites haemorroidalis* larvae, here used as predation risk, is a natural predator of other immature culicids and share the same oviposition habitats as *Ae. aegypti*. Accordingly, *Toxorhynchites* larvae have the potential to be used as a biocontrol agent, especially against disease vector species [37]. The presence of *T. haemorrhoidalis* larvae in our experiment could broadcast to chemical cues that, if detected by the *Ae. aegypti* larvae, could change the adult emergence pattern. Many studies on pest control use predation cues to manage pests. For example, beetle larvae consumed fewer leaves when these leaves were previously exposed to predators [38]. Chemical trails of ladybird on plants repel aphids that are cereal pest [39]. However, our finding suggests that biocontrol strategies based on predation cues are not

effective in the control of *Ae. aegypti* larvae. However, it is important to highlight that it does not mean the direct predation of *Toxorhynchites* is not efficient as a biocontrol agent.

Some non-exclusive explanations can account for the absence of the interaction effect between SCCS and predation risk on *Ae. aegypti* larvae survivorship and adult emergence pattern. This absence may be due to an alteration caused by climatic variables in the release of chemical cues or decreasing prey sensitivity to chemical signals of predator presence [16, 40]. In addition, the presence of *Toxorhynchites* does not seem to be a real threat to some mosquito species. For example, species of *Aedes* select sites for oviposition based on levels of available organic matter, and they do not avoid areas where predators are present. As a result, females prioritize sites with abundant resources for their progeny, regardless of predator presence [41, 42]. Although the ability to detect predation risk varies among Diptera, Romero et al. [43] observed that predation risk did not affect Diptera flower visitation rates, contrary to many other insect orders assessed by them. Thus, as with other dipterous insects, *Ae. aegypti* larvae may not be able to detect predation risk.

Our results revealed that predation risk does not modify *Ae. aegypti* adult emergence patterns, but also showed that an increase in climatic variables (temperature and $CO_2$ concentration) caused distinct effects on emergence distribution of *Ae. aegypti*. We observed that the Intermediate and not the Extreme SCCS sped up the emergence of adult *Ae. aegypti*, agreeing with previous studies showing that warmer environments increase development [19–21]. Typically, the relationship between temperature and life history traits is non-linear [44], and species have a thermal optimum to complete their development. Temperatures above the thermal optimum decrease species performance (e.g. immature survival). Although emergencies started earlier in the Extreme SCCS, the Intermediate SCCS revealed a pronounced emergence on a single day (10th day). This suggests that the larval response to predation risk does not change the phenological patterns of *Ae. aegypti* larval development, but reveals that individual SCCS can alter patterns of emergence, a result with consequences and implications for human health.

Considering that *Ae. aegypti* is the vector of several diseases, and that the Intermediate SCCS is not so far from becoming a reality given the nature of mitigation measures taking place and the speed of their implementation, it is important to carry out this species might behave in the face of SCCS. As the climate change patterns across Amazonia will not be homogeneous [45], it is important to carried out such experiments in ways that incorporate the nature of regional differences. There are many uncertainties in how combined effects of biotic and abiotic factors may influence *Ae. aegypti* larval life-history characteristics; our results add new pieces to this puzzle. Western Amazonia and regions with similar climatic conditions, will probably suffer increases in mosquito populations, partly as a result of the intensive urbanization process (the main driver to their establishment), and partly as a result of climate change, since, as we showed here, *Ae. aegypti* larvae develops faster under SCCS. Our study showed that biocontrol methods simulating predation risk using *T. haemorrhoidalis* larvae are unlikely to be effective for *Ae. aegypti* control, because these signals have no effect on this species. Though the use of *T. haemorrhoidalis* in direct biological control should not be discounted. Therefore, in the near future, shorter life cycles will result in high numbers of mosquitoes, with potential increase in cases of diseases caused by this vector.

## Supporting information

**S1 Fig. Mean of daily temperature in each of the four simulated climate change scenarios during fourteen days of the experiment.**
(TIF)

**S2 Fig. Mean of daily $CO_2$ concentration in each of the four simulated climate change scenarios during fourteen days of the experiment.**
(TIF)

**S3 Fig. Diagnostic plots for the ANOVA model of survival as a function of predator (presence and absence) and SCCS (control, light, intermediate and extreme).**
(TIF)

**S4 Fig. Diagnostic plots for the ANOVA model of emergence skewness as a function of predator (presence and absence) and SCCS (control, light, intermediate and extreme).**
(TIF)

**S5 Fig. Diagnostic plots for the ANOVA model of emergence kurtosis as a function of predator (presence and absence) and SCCS (control, light, intermediate and extreme).**
(TIF)

## Acknowledgments

We are grateful to Mohamed F. Sallam, Rachel Sippy, and two other reviewers for their useful comments and advice. We also thank Dr. Adalberto Val for microcosm use, MSc. Raquel Telles de Moreira Sampaio and Ulysses Barbosa for creation rooms of *Toxorhynchites*, and MSc. William Ribeiro da Silva for support in mosquitoes breeding. Universidade Federal de Mato Grosso do Sul (UFMS) and Instituto Nacional de Pesquisas da Amazônia (INPA) for the support in experiments.

## Author Contributions

**Conceptualization:** Ana C. Piovezan-Borges, Francisco Valente-Neto, Fabio O. Roque.

**Formal analysis:** Ana C. Piovezan-Borges, Francisco Valente-Neto.

**Funding acquisition:** Ana C. Piovezan-Borges.

**Investigation:** Ana C. Piovezan-Borges.

**Resources:** Wanderli P. Tadei, Neusa Hamada.

**Writing – original draft:** Ana C. Piovezan-Borges.

**Writing – review & editing:** Ana C. Piovezan-Borges, Francisco Valente-Neto, Neusa Hamada, Fabio O. Roque.

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
