## [Decision Letter · Decision Letter 0]

29 May 2020

PONE-D-20-07577

Climate change accelerates the emergence of Aedes aegypti, but the outcome is not boosted by predation risk

PLOS ONE

Dear Dr. Piovezan Borges,

Thank you for submitting your manuscript to PLOS ONE. After careful consideration, we feel that it has merit but does not fully meet PLOS ONE’s publication criteria as it currently stands. Therefore, we invite you to submit a revised version of the manuscript that addresses the points raised during the review process.

We look forward to receiving your revised manuscript.

Kind regards,

Jiang-Shiou Hwang, Ph.D.

Academic Editor

PLOS ONE

Journal Requirements:

2. In your Methods section, please provide additional location information of the collection sites, including geographic coordinates for the data set if available.

3. In your Methods section, please provide additional information regarding the permits you obtained for the work. Please ensure you have included the full name of the authority that approved the collection sites access and, if no permits were required, a brief statement explaining why.

Reviewers' comments:

Reviewer's Responses to Questions

**Comments to the Author**

1. Is the manuscript technically sound, and do the data support the conclusions?

Reviewer #1: Partly

Reviewer #2: Partly

2. Has the statistical analysis been performed appropriately and rigorously? 

Reviewer #1: No

Reviewer #2: Yes

3. Have the authors made all data underlying the findings in their manuscript fully available?

Reviewer #1: Yes

Reviewer #2: Yes

4. Is the manuscript presented in an intelligible fashion and written in standard English?

Reviewer #1: No

Reviewer #2: No

5. Review Comments to the Author

Reviewer #1: Climate change accelerates the emergence of 1 Aedes aegypti, but the outcome is not boosted by predation risk

First, thanks for taking the chance to read these interesting results presented by authors. Authors hypotheses was that climate change scenarios in combination with predation risk affect the development rate of Aedes aegypti in terms of adult emergence and larval survivorship of Ae. aegypti. Although the study is well designed and I liked the statistical approach, the hypothesis was unclearly stated in the introduction section and need to be rephrased. Additionally, the statistical analysis needs to be revisited because the two-way ANOVA test was not designed to measure the interaction of two stressors and the combined effect of both on the prey emergence and survivorship. In ANOVA, usually the impact of each predictor on the prey number is evaluated separately. The best approach to measure the combined effect is either PCA or regression analysis. Although authors used linear mixed regression analysis, but it was not clear if it was a preliminary step before two-way ANOVA or a separate analysis.

The entire manuscript needs to be revised by native English speaker to resolve the confusion in many parts in introduction and discussion sections. The hypothesis needs to be clearly stated in the introduction and not confusing between terms such as density, survivorship, development, metamorphosis, and oviposition.

Throughout the manuscript, authors tended to confuse between larval density and survivorship. Although survivorship rate is an indicator for density, the survivorship of larvae in the current study design was due to climate change scenario rather than predation risk because larvae were not exposed directly to their predator. It would be better if authors be more specific and not use larval density as synonym of survivorship or emergence.

In the discussion section, there were lots of conjectures and general conclusion about mosquito development without direct relevance to their findings. Authors need to be more specific in discussing their findings with similar studies in the light of the current findings in their study. I overlaid that line by line in the specific comments.

Specific comments

Abbreviation of all genus names of mosquito need to be properly written as Ae. and not A.

Line 21: change "predation-prey" to "predator-prey"

Line 28: replace "to" with "and"

Line 36: What is "than" here for? Delete it

Line 108-110: Very long sentence. Needs to be concise.

Line 111-116: What is that section? is this a result or introduction? It sounds line it is results. Needs to be rephrased.

Line 112: Lethal levels

Line 113: Using the term density here is not appropriate and I suggest being deleted. Authors mentioned that they predicted the influence of predation risk on prey density; however, in the current design they only used fixed number of larvae per each replicate. When density is mentioned, readers would get the impression of lethal effect caused by predation. I'm not sure if that would be a good indicator for the impact of predation risk on prey’s density, especially Ae. aegytppi larvae were not exposed directly to the predator, instead, predator was fed with an extra larva every day. Therefore, how come this would affect the larval density in each replicate. Based to the current experimental design, emergence rate and survivorship of Ae. aegypti is a function of climate change scenarios and not the predation risk.

Line 114-116: This paragraph sounds like the authors were trying to state their hypothesis. They need to rephrase the sentence to make it sounds like it a legit hypothesis and not results. The paragraph sounds like it is a results section, or discussion.

I believe authors meant to say they hypothesized that the effect of both stressors, in combination, would affect adult emergence and survivorship.

In addition, it was not quite clear how authors determined the survivorship of the larvae for both predator and prey? This needs to be elaborated, especially the larvae were not exposed directly to their predator.

Line 154: Is this insecticide resistant or susceptible strain? Please elaborate.

Line 161: I'm not sure the factorial analyses approach is the right approach to study the combined effect of climate change scenarios and predation risk on mosquito emergence and survivorship.

The two-way ANOVA test is a very good stat approach; however, it was not designed to measure the interaction of two predictors and the combined effect of both on the prey emergence and survivorship. In ANOVA, usually the impact of each predictor on the prey number is evaluated separately. The best approach to measure the combined effect is either PCA or regression analysis.

Line 185-187: This 4th instar larvae from outside and not from the same container, right?

It is not clear if the predator emerged before their preys or how the authors managed that.

Line 191: what was the male: female ratio? I’m not surprised that males emerged before females, especially under adverse conditions. However, authors did not mention what was the male : female ratio? Also, more clarification is needed on this issue in the results and discussion sections.

Line 265: Do you mean female biomass was larger than male? Need to be clarified.

Line 275-279: I'm not surprised with this conclusion. Many of previous studies evaluated the geographic expansion of Ae. aegypti in response to climate change at both spatial and temporal scales. What is the relevance of this sentence to the current study?

Line 281-285: Again, what is the take home message from this sentence? What authors are trying to say here? and what is the relevance of this sentence to the current study?

Line 287: Use development instead of metamorphosis

Line 288: Use larval instead of larvae

Line 290: Although it is a general statement and well proven in previous studies, which is correct to certain limit, authors did not support this conclusion in their results. In Fig 2, the number of survivorships in the control replicates are decreasing with the increase in temperature and CO2 concentration.

Additionally, there are other factors in the environment that might affect the development rates and survivorship of this mosquito species. For example, hot and dry weather will not be a suitable condition for the survivability of this mosquito. Additionally, presence of other competitors such as Ae albopictus is another big factor that limit the spatial and temporal distribution of Ae aegypti.

I would suggest authors to be specific in their conclusion and not to generalize it.

Line 292: The current study handled only the larval stages. I would suggest the authors to use different term like larval development.

Line 292-294: This sentence is irrelevant to the current study. What is the main message here?

Line 297-302: The current study did not address oviposition behavior! What is the whole point of mentioning that here?

Line 311-316: This section is results and does not belong to discussion. Suggest rephrasing or move it to results.

Line 319-327: What is the relevance of this to the current study?

Line 333: I would suggest replacing "life-history" with "development".

Line 336-337: This conclusion stated by authors was not supported in their results. Besides, Aedes aegypti is a well-known species that only occur at truly urban areas. Accordingly, the increase in urban settings in Amazonia is the direct cause of the establishment of this mosquito species.

It was kind of intriguing that the survivorship rates in figure 2 are smaller than emerged adults in figure 5 for their corresponding treatments. For example, in figure 2, the number of survivorship larvae in control predation risk was almost 92% (=55 larvae). Meanwhile, in figure 5, the number of emerged adults of Ae. aegypti was 57 for the same experiment. Similarly, the number of survivorship larvae in light climate change scenario with predation risk was almost 97% (=58 larvae). Meanwhile, in figure 5, the number of emerged adults of Ae. aegypti was 60 for the same experiment. That being said, and according to the figure 2 and 5, predation risk increased the survivorship rates and emerged adults of mosquito in predation experiments compared to their control. However, no significant difference was reported.

I think, it will be better if authors used emergence rate instead of numbers for the sake of consistency and transparency of data visualization and representation. Same thing for the other two scenarios. With that being

Figure 2 and 5, needs to be consistent in terms of either using rate or number. Standard errors in replicates are too big, do not you think that might be the reason of having insignificant differences between replicates.

Font size in heading and subheadings need to be fixed based on the journal’s guidelines.

Reviewer #2: Review of PONE-D-20-07577

In this article, the authors use a microcosm experimental set up, to simulate climate change scenarios for larval habitat of Aedes aegypti mosquitoes, manipulating temperature and CO2 levels, and introducing a larval predator into replicates, to explore the impact on emergence time and larval survivorship to emergence both with and without predation.

This is a fairly straightforward experimental design, and the authors found that the simulated climate change scenarios sped up emergence, as expected, but larval survivorship was not affected by the simulated climate change scenarios, nor by the presence of predation in the replicates.

I found the writing to be good, but the story muddled, and therefore in need of some clarification and reframing. Key among this is that they need to be very clear that this is a location specific study, simulating future scenarios in that specific location (Manaus), and that microcosm experiments of climate changes on larval Aedes aegypti are not new, so this is not novel information for managing human health, as they currently seem to be claiming.

The novelty is in looking at the role of predation under these warming scenarios, and the fact that it does not change in impact is important for biocontrol of larval Aedes – knowing that a larval predator is not having a measurable impact on the emergence acceleration IS useful.

General Comments:

I would recommend establishing an acronym Simulated Climate Change Scenario (SCCS) early on, and using it where ‘climate change’ is casually used – this will disambiguate what is being tested, versus discussion of the literature. Be very clear in the abstract and introduction exactly what is being simulated – it is the climate of 2100, in terms of temperature and CO2 concentration, projected under specific scenarios.

At first mention, use Aedes aegypti (italicized), then put Ae. aegypti in parentheses to clarify how you’ll refer to it there onward – then do that. If you are referring to Aedes spp (as in more than one species), make it clear with spp.

Be very clear that this is a larval experiment – it sounds like adults in a lot of the text, and the naïve reader won’t know what you are doing. This is the aquatic phase, and at emergence, the experiment ends. I would add ‘larval’ to every mention of the Ae. aegypti in the experiment when you describe it (intro through discussion).

Title: I suggest changing it to:

Simulated climate change accelerates Aedes aegypti emergence, but not larval predation impact, in a microcosm experiment in Manaus, Brazil.

Abstract

Line 21: predator-prey

Lines 23-25 – clarify that this is climate change simulated for Manaus in 2100

Lines 25-26 – given it says you followed the 2007 IPCC report, the RCPs for the scenarios should be given explicitly, and which version of the projections were used? Is this AR4 AR5? What are ‘light’ ‘intermediate’ and ‘extreme’??

Line 32 – Neither simulated climate change scenarios nor predation risk affected Ae. aegypti…

Line 35 - …emergence pattern of

Line 38 – synchronously

Line 39-40 – clarify that this is climate change in Manaus – Ae aegypti are NOT resistant to simulated climate change everywhere – there are places that will be too hot even for aegyptis.

Line 44 – since this is already established and known, perhaps switch the message to say ‘in Manaus’ and to emphasize the lack of impact of a potential larval biocontrol agent.

Introduction

Firstly, I would take Line 281-290 out of the discussion, and make them the opening of the Introduction. This clearly sets the scene for the whole framework of the experiment, and why we care about larval stages when we know so much about the adult stages.

Line 58 – a great place to emphasize that most studies are based on adult stages because that is the transmitting stage, but we know less about impacts on larval life-history, which will carry over. Again, a great chance to mention larval biocontrol methods, and a need to understand them better.

Lines 98-104 – name the explicit climate scenarios and versions (as I mentioned for the abstract) and then refer to these explicitly throughout. Be really clear that these are specifically tuned to Manaus for this microcosm experiment. Location is everything.

Line 105 – this sentence seems to be odd – I think you mean in your specific scenarios only, in which case, start with: In Manaus…

Line 110: when you mention survivorship, it sounds like you mean for adult mosquitoes if you don’t add in ‘larval’. So far, there has been no mention of the methods about measuring up to emergence.

Lines 117-119: Again, this is already a known quantity – you are not setting off alarms, decision makers already know that life-history of Aedes speeds up with climate change. What you are indicating is that there doesn’t seem to be predation impact changes, which calls for more investigation.

Methods

Climate change scenarios – same comment – which ones, make it clear it’s in Manaus, and line 129 – they simulate the year 2100, you are not ramping up temperature and CO2 over 80 years. Which RCPs? Which GCMs? Which iteration of the IPCC framings?

Lines 143-144 – this is the first time I really understood that this was a larval predation experiment – perhaps see if you can make that more obvious much earlier in the paper.

Discussion (I got tired of doing line edits, sorry)

Perhaps open with a recap of the framework – as climate change causes life history of Ae aegypti to accelerate, so we expect to see increased numbers of disease cases due to increased bites; we found that the rate of emergence accelerated under simulated climate change and larval survivorship was not impacted, and neither of these were affected by predation. This has implications for human health, as the impact of larval biocontrol may not keep pace with life-history acceleration under climate change.

Line 268: ‘larval’ survivorship

Lines 276-277 – this is already known, it’s the lack of evidence that larval predation changes that is interesting.

Line 297 - … of Ae aegypti, under the different simulated scenarios, which corroborates findings in previous, non-climate-change studies.

Lines 337-340 – THIS IS KEY – this is the most essential piece of finding in this study – make sure this is highlighted, emphasized, underscored, etc, right from the beginning.

6. PLOS authors have the option to publish the peer review history of their article (what does this mean?). If published, this will include your full peer review and any attached files.

Reviewer #1: No

Reviewer #2: No

---

## [Author Response · Author response to Decision Letter 0]

15 Jul 2020

Reviewer 1:

Comment: First, thanks for taking the chance to read these interesting results presented by authors. Authors hypotheses was that climate change scenarios in combination with predation risk affect the development rate of Aedes aegypti in terms of adult emergence and larval survivorship of Ae. aegypti. Although the study is well designed and I liked the statistical approach, the hypothesis was unclearly stated in the introduction section and need to be rephrased…

The entire manuscript needs to be revised by native English speaker to resolve the confusion in many parts in introduction and discussion sections. The hypothesis needs to be clearly stated in the introduction and not confusing between terms such as density, survivorship, development, metamorphosis, and oviposition. 

Response: Thank you for your positive comment. The hypothesis has been rewritten. In this version we clarify our expectation concerning the effects of predation risk on the emergence pattern of adult Aedes aegypti. As suggested, the manuscript was revised by a native English speaker, Prof. Adrian Barnett.

Comment: Additionally, the statistical analysis needs to be revisited because the two-way ANOVA test was not designed to measure the interaction of two stressors and the combined effect of both on the prey emergence and survivorship. In ANOVA, usually the impact of each predictor on the prey number is evaluated separately. The best approach to measure the combined effect is either PCA or regression analysis. Although authors used linear mixed regression analysis, but it was not clear if it was a preliminary step before two-way ANOVA or a separate analysis.

Response: Thank you for your comment. We believe that two-way ANOVA is suited to test our hypothesis and to use in our factorial experimental design. First, we recognize that ANOVA is not suited to test synergism between predictors or the combined effect of both predictors as we stated in the first version of the manuscript, but two-way ANOVA allows us to test the interaction between both predators and simulated climate change scenario (SCCS). This interaction term (predation* SCCS) measures differences in the treatment group means that cannot be predicted on an additive basis by the two main effects (predation or SCCS). In other words, it estimates the effect of one factor (e.g. SCCS) that depends on the level of the other factor (e.g. predation). Quantifying the interaction term is the key reason to have carried out a factorial study. So, following your suggestion, we reviewed the introduction considering that ANOVA cannot detect synergisms or combined effects, only interaction between predictors.

Second, ANOVA is a linear model designed to model relationship between a continuous dependent variable and categorical predictors. Its aim is to compare means between groups that were randomly sampled. Regression analysis requires continuous response and predictor variables. Both ANOVA and linear regression are special cases of generalized linear models; indeed we could use another generalized linear model with appropriate distribution, but our models met Gaussian linear models assumptions. As you can see from the data, the diagnostic plots show homogeneity of variances and normality (Fig 1 for survival; Fig 2 for emergence skewness; Fig 3 for emergence kurtosis). These plots are also included in Supporting Information (S3-S5 Figs).

Sorry, but we did not completely understand your recommendation regarding the use of PCA. PCA reduces data dimensionality, but cannot be used to test hypothesis. We believe that your suggestion was to use PCA to reduce dimensionality of the number of individuals that emerged per day (starting on the fourth day of experiment and ending on the fourteenth day) instead of using kurtosis and skewness. Accordingly, we carried out a PCA analysis with number of emerged individuals emerging per day (see figure below), and the two PCA axes captured 87% of the data variation. PCA axis 1 separated mainly control and light replicates from intermediate and extreme SCCS, irrespective of predator presence or absence. PCA axis 2 is more difficult to interpret, but seems to separate light and intermediate replicates from extreme SCCS. However, we still believe that kurtosis and skewness are more straightforward measures, because they measure asymmetry in data distribution. They allowed us to detect the peak of emergence, which is something we could not do with PCA (Figure is in the file "Response to Reviewers"). Accordingly, we retained the two-way ANOVA for survival, emergence skewness and kurtosis.

Finally, we used a linear mixed model (LMM) (with Gaussian distribution) in a separate analysis to analyze emergence, i.e., it did not act as a preliminary step preceding the two-way ANOVA for distribution of emergence time (measures by kurtosis and skewness). In this model we considered the replicates as a random factor (replicates were our repeated measure), with an autocorrelation function to control repetition across days, and SCCS and predation risk as fixed factors. This model is equivalent to a two-way ANOVA with repeated measures. LMM analysis was used to detect if predation and SCCS and their interaction affect number of emergences, controlling for repeated measures and time autocorrelation. The model output is not informative in terms of temporal patterning, because we did not include time as a fixed factor, so as to retain a viable number of degrees of freedom. To better understand this effect of time without including time in our models as fixed factor, we preferred to use the approach mentioned above (calculate kurtosis and skewness of distribution of emergence time). 

Comment: Throughout the manuscript, authors tended to confuse between larval 

density and survivorship. Although survivorship rate is an indicator for density, the survivorship of larvae in the current study design was due to climate change scenario rather than predation risk because larvae were not exposed directly to their predator. It would be better if authors be more specific and not use larval density as synonym of survivorship or emergence.

Response: Thank you for your comment. We agree that these terms were confused in some parts of the text. We specify the terms using “larval survivorship” and “adult emergence pattern”, we revised these terms throughout the text and have made corrections in the new version.

Comment: In the discussion section, there were lots of conjectures and general conclusion about mosquito development without direct relevance to their findings. Authors need to be more specific in discussing their findings with similar studies in the light of the current findings in their study. I overlaid that line by line in the specific comments.

Response: In the new version we highlight the results concerning predation risk. As this is the central novelty of our study, we inserted some ideas on the use of predation risk cues as Ae. aegypti biocontrol strategy. As a result, we removed some general sentences and improved the discussion of our main findings.

The specific comments that were highlighted line by line in the discussion are answered individually below.

Specific comments

Comment: Abbreviation of all genus names of mosquito need to be properly written as Ae. and not A.

Response: Thank you for your observation. The genus names were corrected throughout the text. 

Comment: Line 21: change "predation-prey" to "predator-prey"

Response: Changed.

Comment: Line 28: replace "to" with "and"

Response: Changed.

Comment: Line 36: What is "than" here for? Delete it

Response: Corrected.

Comment: Line 108-110: Very long sentence. Needs to be concise.

Response: The sentence was rewritten, and in the new version we clarify our overall goal that is to gain an understanding single or interaction effects of SCCS and predation risk in larval survivorship and adult emergence pattern of Ae. aegypti. 

Comment: Line 111-116: What is that section? is this a result or introduction? It sounds line it is results. Needs to be rephrased.

Response: Thank you for your comment. The hypothesis has been rewritten. This information is in at lines 122-128.

Comment: Line 112: Lethal levels

Response: Corrected.

Comment: Line 113: Using the term density here is not appropriate and I suggest being deleted. Authors mentioned that they predicted the influence of predation risk on prey density; however, in the current design they only used fixed number of larvae per each replicate. When density is mentioned, readers would get the impression of lethal effect caused by predation. I'm not sure if that would be a good indicator for the impact of predation risk on prey’s density, especially Ae. aegypti larvae were not exposed directly to the predator, instead, predator was fed with an extra larva every day. Therefore, how come this would affect the larval density in each replicate. Based to the current experimental design, emergence rate and survivorship of Ae. aegypti is a function of climate change scenarios and not the predation risk.

Response: We agreed with your comment and we rephrased the sentence to make it clear that our hypothesis does not predict the effect of SCCS and predation risk on Ae. aegypti larval survivorship.

Comment: Line 114-116: This paragraph sounds like the authors were trying to state their hypothesis. They need to rephrase the sentence to make it sounds like it a legit hypothesis and not results. The paragraph sounds like it is a results section, or discussion.

I believe authors meant to say they hypothesized that the effect of both stressors, in combination, would affect adult emergence and survivorship.

Response: Thanks for your observation, we rephrased the sentence as suggest. The sentence is in at lines 122-128.

Comment: Line 114-116: …In addition, it was not quite clear how authors determined the survivorship of the larvae for both predator and prey? This needs to be elaborated, especially the larvae were not exposed directly to their predator.

Response: We determined larval survivorship only for prey. At the start of our experiment we added a fixed number of Ae. aegypti first instar larvae per replicate (n=60). Then, to determine survivorship we divided the final number of emerged adults in each replicate by 60. We explained this in the new version, at lines 176-178. 

Comment: Line 154: Is this insecticide resistant or susceptible strain? Please elaborate.

Response: No, it is not an insecticide resistant or susceptible strain. The Ae. aegypti eggs used in the experiment are from colonies established in laboratory from wild-caught eggs collected in Manaus. We added this essential detail in Material and Methods section, as follows:

“Eggs of Ae. aegypti were obtained from colonies held by the Malaria and Dengue Laboratory at INPA. These colonies were established with wild-caught eggs collected in Manaus, using oviposition traps. Aedes aegypti eggs were collected with authorization and approval of the Brazilian Biodiversity Authorization and Information System (SISBIO; Permit 61563). We placed filter paper to collect eggs of 4th and 5th generation adults from these colonies, and used these in the experiment. In each SCCS, the filter paper containing the eggs were placed in plastic containers until hatching occurred (± 17 hours)”.

Comment: Line 161: I'm not sure the factorial analyses approach is the right approach to study the combined effect of climate change scenarios and predation risk on mosquito emergence and survivorship.

The two-way ANOVA test is a very good stat approach; however, it was not designed to measure the interaction of two predictors and the combined effect of both on the prey emergence and survivorship. In ANOVA, usually the impact of each predictor on the prey number is evaluated separately. The best approach to measure the combined effect is either PCA or regression analysis.

Response: Thanks again for your suggestion. Responses to your second comment have been incorporated into our statistical approach explanation above.

Comment: Line 185-187: This 4th instar larvae from outside and not from the same container, right? 

Response: Yes, it is. We add this information in text, at line 202.

Comment: Line 185-187: …It is not clear if the predator emerged before their preys or how the authors managed that. 

Response: The life cycle of the predator T. haemorrhoidalis is longer than that of Ae. aegypti life cycle. The larval stage of T. haemorrhoidalis lasts for at least 20 days, depending on the temperature and food conditions under which they exist. Thus, the predator remained in the larval stage throughout the experiment. 

Comment: Line 191: what was the male: female ratio? I’m not surprised that males emerged before females, especially under adverse conditions. However, authors did not mention what was the male : female ratio? Also, more clarification is needed on this issue in the results and discussion sections.

Response: After your comment about possible data discrepancy between Fig. 2 and 5, we checked all data and found inconsistency in the data used in Fig. S3 (daily emergence of males and females) and the data used in Fig. 2 and 5. This inconsistency occurred because, during the aspiration of mosquitoes in each replicates within the simulated climate change scenarios (microcosm), some individuals escaped. These individuals were counted for total survival and emergence, but they were not captured and, consequently, we cannot determine their sex. This happened mainly because of the brief time spent inside each microcosm room (on average, four minutes). This procedure is taken to avoid sudden changes in rooms conditions due to the manipulator's breathing, which can influence the variables carefully controlled (CO2). As such, it is designed to avoid any influences on the experiments. Even though it is a closed environment, when individuals escape, we cannot spend a lot of time for this purpose because of the strict rules in the ADAPTA. Therefore, these individuals were included in the total count, but not in the count of males and females, creating this discrepancy that was observed in the data.

Our aim did not involve the assessment of the effect of SCCS and predation risk on male and female of Ae. aegypti, instead, we central aim was to understand overall Ae. aegypti responses. Besides that, as you can see in the first version, we did not analyze males and females data, we just plotted their patterns and included general comments in the results and discussion, which did not add relevant information and were not the main focus of our research. Therefore, considering the issues addressed above, all sentences throughout the text concerning males and females, the biomass table and the figure with daily emergence of females and males that were in the supporting information were removed. In this way, we keep throughout the text the essential data, corresponding to the objective and associated hypothesis as initially proposed. 

Comment: Line 265: Do you mean female biomass was larger than male? Need to be clarified.

Response: As explained in the comment above (line 191), the data and sentences about males and females have been removed from the text.

Comment: Lines 275-279: I'm not surprised with this conclusion. Many of previous studies evaluated the geographic expansion of Ae. aegypti in response to climate change at both spatial and temporal scales. What is the relevance of this sentence to the current study?

Response: This sentence was removed from the Discussion section. In the new version, we focus on more specific sentences that link with our results and not on general sentences as occurred it was in the first version.

Comment: Line 281-285: Again, what is the take home message from this sentence? What authors are trying to say here? and what is the relevance of this sentence to the current study?

Response: We agreed with your comment. This sentence is not relevant, so we have removed it from the Discussion section.

Comment: Line 287: Use development instead of metamorphosis

Response: Corrected.

Comment: Line 288: Use larval instead of larvae

Response: Corrected.

Comment: Line 290: Although it is a general statement and well proven in previous studies, which is correct to certain limit, authors did not support this conclusion in their results. 

Response: This was removed. The sentence was rewritten, a better explanation now present is in the penultimate paragraph of the Discussion section. We supported this affirmation with results that can be seen in Fig. 5. In the control SCCS, adults being on day 9, while for extreme SCCS this starts on day 7, so for extreme SCCS the emergence starts earlier, this explanation is also present in the penultimate paragraph of the Discussion section.

Comment: Line 290: …In Fig 2, the number of survivorships in the control replicates are decreasing with the increase in temperature and CO2 concentration.

Response: Although the mean survival value for the treatment with predation risk in the control scenario was the lowest among all scenarios, when we compared the survivorship between SCCS and predation risk, statistical analysis does not indicate a significant difference (data described in the Results). Note that the bars in Fig 2 are the mean confidence intervals (95%).

Comment: Line 290: … Additionally, there are other factors in the environment that might affect the development rates and survivorship of this mosquito species. For example, hot and dry weather will not be a suitable condition for the survivability of this mosquito. Additionally, presence of other competitors such as Ae albopictus is another big factor that limit the spatial and temporal distribution of Ae aegypti.

I would suggest authors to be specific in their conclusion and not to generalize it.

Response: We agree with your comment, we emphasize that the adult emergence is faster in extreme SCCS, and that high humidity is present in all scenarios. Here we do not specifically address humidity, because the humidity is the same in all scenarios, so it does not influence the obtained results.

Comment: Line 292: The current study handled only the larval stages. I would suggest the authors to use different term like larval development.

Response: We modified the beginning of this sentence. We also modified throughout the text all the expressions that refer to the analyzed variables. Now we always refer to "larval survivorship" and "adult emergence pattern".

Comment: Line 292-294: This sentence is irrelevant to the current study. What is the main message here?

Response: We agreed with your comment. This sentence it is not relevant, so we removed it from the Discussion section.

Comment: Line 297-302: The current study did not address oviposition behavior! What is the whole point of mentioning that here?

Response: We rephrased this sentence, and we simply mention how this species, even in natural habitats, seem not to perceive predator presence, supporting the preceding idea (see lines 316-319).

Comment: Line 311-316: This section is results and does not belong to discussion. Suggest rephrasing or move it to results.

Response: As explained in the comment “line 191”, the sentences concerning males and females have been removed from the text.

Comment: Line 319-327: What is the relevance of this to the current study?

Response: This discussion is not within the proposed objectives, so, as explained in the comment “line 191”, the sentences about males and females have been removed from the text.

Comment: Line 333: I would suggest replacing "life-history" with "development".

Response: Thank you for your suggestion. We added in the Introduction “life-history characteristics (e.g. survivorship and reproduction)”, so we kept “life-history characteristics” in the last paragraph of the text.

Comment: Line 336-337: This conclusion stated by authors was not supported in their results. Besides, Aedes aegypti is a well-known species that only occur at truly urban areas. Accordingly, the increase in urban settings in Amazonia is the direct cause of the establishment of this mosquito species.

Response: We added this important information about urbanization in western Amazonia, specifying the study localization, and we explained that in addition to this driver, climate change will result in the increase of Ae. aegypti population due to their faster cycle compared to the control.

Comment: Line 336-337: …It was kind of intriguing that the survivorship rates in figure 2 are smaller than emerged adults in figure 5 for their corresponding treatments. For example, in figure 2, the number of survivorship larvae in control predation risk was almost 92% (=55 larvae). Meanwhile, in figure 5, the number of emerged adults of Ae. aegypti was 57 for the same experiment. Similarly, the number of survivorship larvae in light climate change scenario with predation risk was almost 97% (=58 larvae). Meanwhile, in figure 5, the number of emerged adults of Ae. aegypti was 60 for the same experiment. That being said, and according to the figure 2 and 5, predation risk increased the survivorship rates and emerged adults of mosquito in predation experiments compared to their control. However, no significant difference was reported.

I think, it will be better if authors used emergence rate instead of numbers for the sake of consistency and transparency of data visualization and representation. Same thing for the other two scenarios. With that being Figure 2 and 5, needs to be consistent in terms of either using rate or number. Standard errors in replicates are too big, do not you think that might be the reason of having insignificant differences between replicates…

Response: Thank you for this comment. As we said above in the line 191 comment, your comment was essential to detect inconsistencies between total survivorship and emergence data and males-females data. We checked all the data to verify if there was any discrepancy between total survivorship and emergence data, and no inconsistencies were found (the raw data are in the table in the file "Response to Reviewers"). 

In figure 2 we showed a mean of survivorship and the mean confidence interval (95%), while in figure 5, we showed a mean and standard error of the 4 replicates per treatment/day, so it is difficult to make a comparison between survivorship and emergence data just looking the figures. We emphasize that the values of survivorship and emergence used in statistical analysis are exactly the same.

Statistical analysis does not show differences between the two treatments (predation risk and control), so these differences were not reported. As suggested, we modified the y-axis unit in Fig. 5, and it is now reported in percentage, to improve data visualization. Sorry, we put in the legend "mean (SE)" in figures 2, 3 and 4, but, in fact, the bars are the mean confidence interval (95%). We corrected this in the new version.

Comment: Line 336-337: …Font size in heading and subheadings need to be fixed based on the journal’s guidelines.

Response: We checked the font size in heading and subheadings. They were already in accordance with the journal’s guidelines.

Reviewer 2:

Comment: I found the writing to be good, but the story muddled, and therefore in need of some clarification and reframing. Key among this is that they need to be very clear that this is a location specific study, simulating future scenarios in that specific location (Manaus), and that microcosm experiments of climate changes on larval Aedes aegypti are not new, so this is not novel information for managing human health, as they currently seem to be claiming.

The novelty is in looking at the role of predation under these warming scenarios, and the fact that it does not change in impact is important for biocontrol of larval Aedes – knowing that a larval predator is not having a measurable impact on the emergence acceleration IS useful.

Response: We agreed with you on both points, so we specify the study location, and highlight the novelty of the work throughout the text. Each specific comment on these and other items is answered below.

General comments

Comment: I would recommend establishing an acronym Simulated Climate Change Scenario (SCCS) early on, and using it where ‘climate change’ is casually used – this will disambiguate what is being tested, versus discussion of the literature. Be very clear in the abstract and introduction exactly what is being simulated – it is the climate of 2100, in terms of temperature and CO2 concentration, projected under specific scenarios.

Response: Thank you very much for your suggestion. We used the acronym SCCS throughout the text. Whenever possible we added the information that the SCCS are predicted by AR4 IPCC (2007) for the year 2100 in Manaus. 

Comment: At first mention, use Aedes aegypti (italicized), then put Ae. aegypti in parentheses to clarify how you’ll refer to it there onward – then do that. If you are referring to Aedes spp (as in more than one species), make it clear with spp.

Response: Revised.

Comment: Be very clear that this is a larval experiment – it sounds like adults in a lot of the text, and the naïve reader won’t know what you are doing. This is the aquatic phase, and at emergence, the experiment ends. I would add ‘larval’ to every mention of the Ae. aegypti in the experiment when you describe it (intro through discussion).

Response: In this new version, we specified that our experiments were done with Ae. aegypti larvae. For a better understanding, we also quote every time as "larval survivorship" and " adult emergence pattern".

Title

Comment: I suggest changing it to:

Simulated climate change accelerates Aedes aegypti emergence, but not larval predation impact, in a microcosm experiment in Manaus, Brazil.

Response: Thank you for your suggestion. We accepted your suggestion, but we made a small change to keep in the title "predation risk" to avoid any confusion. We did not use direct predation in our study, in which the predator actually consumed prey. We also believe that is better to use “western Amazonia” instead “Manaus” because our results may work for this entire region. We have specified the location in several sections in the text.

Abstract

Comment: Line 21: predator-prey

Response: Corrected.

Comment: Lines 23-25 – clarify that this is climate change simulated for Manaus in 2100

Response: Corrected.

Comment: Lines 25-26 – given it says you followed the 2007 IPCC report, the RCPs for the scenarios should be given explicitly, and which version of the projections were used? Is this AR4 AR5? What are ‘light’ ‘intermediate’ and ‘extreme’??

Response: Thanks for your comment. We detailed this in the abstract (lines 40-41) and in the Methods section (lines 141).

Comment: Line 32 – Neither simulated climate change scenarios nor predation risk affected Ae. aegypti…

Response: Corrected.

Comment: Line 35 - …emergence pattern of

Response: Corrected.

Comment: Line 38 – synchronously

Response: Corrected.

Comment: Line 39-40 – clarify that this is climate change in Manaus – Ae aegypti are NOT resistant to simulated climate change everywhere – there are places that will be too hot even for aegyptis.

Response: Corrected.

Comment: Line 44 – since this is already established and known, perhaps switch the message to say ‘in Manaus’ and to emphasize the lack of impact of a potential larval biocontrol agent.

Response: We followed your suggestion and emphasized that predation risk, or the interaction between predation risk and SCCS, did not impact any of the analyzed variables in Ae. aegypti reared in SCCS in Manaus. We emphasized that effects of SCCS may have implication for human health.

Introduction

Comment: Firstly, I would take Line 281-290 out of the discussion, and make them the opening of the Introduction. This clearly sets the scene for the whole framework of the experiment, and why we care about larval stages when we know so much about the adult stages

Response: Thank you for your suggestion. We removed this paragraph of the Discussion and used its main messages in the Introduction. However, we prefer not to change the Introduction structure.

Comment: Line 58 – a great place to emphasize that most studies are based on adult stages because that is the transmitting stage, but we know less about impacts on larval life-history, which will carry over. Again, a great chance to mention larval biocontrol methods, and a need to understand them better.

Response: Thank you for your comment. As suggested we added this sentence at the end of first Introduction paragraph “However, few studies focus on larval life-history, despite it is well known that changes that occur in the environment in the larval stage, such as climate change, may shape the development and behavior of adults (known as carry-over effects) [8].” We mentioned biocontrol strategies in the penultimate paragraph of the Introduction section.

Comment: Lines 98-104 – name the explicit climate scenarios and versions (as I mentioned for the abstract) and then refer to these explicitly throughout. Be really clear that these are specifically tuned to Manaus for this microcosm experiment. Location is everything.

Response: As suggested previously, we detailed SCCS in the Abstract (lines 35-41) and in the Methods section (lines 139-146). We highlighted that the experiment was developed in Manaus, in the Introduction section we added information about the location, as follow “Accordingly, we conducted an experiment in a microcosm simulating real-time climatic condition in Manaus (Control) and gradual increase in temperature and CO2 in other three SCCS for this city in the year 2100”. 

In addition, as we explained in your comment regarding the title, when we refer to our results, we used "western Amazonia" because our results may be applicable across the entirety of this region.

Comment: Line 105 – this sentence seems to be odd – I think you mean in your specific scenarios only, in which case, start with: In Manaus…

Response: We changed this sentence to “It is widely known that temperature increases, within thermal tolerance, affects development and behavior of Ae. aegypti”

Comment: Line 110: when you mention survivorship, it sounds like you mean for adult mosquitoes if you don’t add in ‘larval’. So far, there has been no mention of the methods about measuring up to emergence.

Response: Throughout the text, we modified all mentions of survivorship and emergence to "larval survivorship" and "adult emergence pattern".

Comment: Lines 117-119: Again, this is already a known quantity – you are not setting off alarms, decision makers already know that life-history of Aedes speeds up with climate change. What you are indicating is that there doesn’t seem to be predation impact changes, which calls for more investigation.

Response: We changed the end of the Introduction, according to your suggestion, and we highlighted that our study brought new results concerning the impacts of predation risk on larval survivorship and time to adult emergence of Ae. aegypti reared under different SCCS.

Methods

Comment: Climate change scenarios – same comment – which ones, make it clear it’s in Manaus, and line 129 – they simulate the year 2100, you are not ramping up temperature and CO2 over 80 years. Which RCPs? Which GCMs? Which iteration of the IPCC framings?

Response: Whenever possible, we have made clear that the experiment was done in Manaus. We reformulated the sentence explained that these simulations are predicted for the year 2100.

In the AR4-IPCC 2007, which was used as a basis for the construction of the microcosm, the scenarios proposed by IPCC were B1, A1B and A2. Then, we maintained the same names used in this report and we relate the microcosm scenarios to the scenarios provided by IPCC 2007.

Comment: Lines 143-144 – this is the first time I really understood that this was a larval predation experiment – perhaps see if you can make that more obvious much earlier in the paper.

Response: Sorry about that. We have restructured the statements regarding effects of predation risk on biodiversity in the Introduction.

Discussion

Comment: Perhaps open with a recap of the framework – as climate change causes life history of Ae aegypti to accelerate, so we expect to see increased numbers of disease cases due to increased bites; we found that the rate of emergence accelerated under simulated climate change and larval survivorship was not impacted, and neither of these were affected by predation. This has implications for human health, as the impact of larval biocontrol may not keep pace with life-history acceleration under climate change.

Response: Thank you for your suggestion. As suggested, we opened the Discussion section with known effects of SCCS on Ae. aegypti, then we added our findings. However, we have also added information about the implications of these results to human health in the following paragraph.

Comment: Line 268: ‘larval’ survivorship

Response: We removed the sentence containing this term. According to our response to comment concerning "line 110", we modified all mentions of survivorship to "larval survivorship" throughout the text.

Comment: Lines 276-277 – this is already known, it’s the lack of evidence that larval predation changes that is interesting. 

Response: Thank you for your comment. We removed this sentence. As you suggested, in the new version we have highlighted the results relating to predation risk. As this is the main novelty of our study, we inserted some ideas on the use of the predation risk cues as a biocontrol strategy of Ae. aegypti.

Comment: Line 297 - … of Ae aegypti, under the different simulated scenarios, which corroborates findings in previous, non-climate-change studies.

Response: We rephrased this sentence to cover the results of other studies (non-climate-change studies), we expanded this discussion and we cited authors that have investigated effects of predation risk on Ae. aegypti. In the new version, this information opens the second paragraph in Discussion section.

Comment: Lines 337-340 – THIS IS KEY – this is the most essential piece of finding in this study – make sure this is highlighted, emphasized, underscored, etc, right from the beginning.

Response: Thank you for your comment. Following to your suggestion re highlighting predation risk responses, we focused on this important outcome, and added information on this in Introduction. Additionally, we modified the Discussion bringing this new approach to this section.

---

## [Decision Letter · Decision Letter 1]

11 Sep 2020

PONE-D-20-07577R1

Simulated climate change, but not predation risk, accelerates Aedes aegypti emergence in a microcosm experiment in western Amazonia

PLOS ONE

Dear Dr. Piovezan Borges,

Thank you for submitting your manuscript to PLOS ONE. After careful consideration, we feel that it has merit but does not fully meet PLOS ONE’s publication criteria as it currently stands. Therefore, we invite you to submit a revised version of the manuscript that addresses the points raised during the review process.

We look forward to receiving your revised manuscript.

Kind regards,

Jiang-Shiou Hwang, Ph.D.

Academic Editor

PLOS ONE

Reviewers' comments:

Reviewer's Responses to Questions

**Comments to the Author**

1. If the authors have adequately addressed your comments raised in a previous round of review and you feel that this manuscript is now acceptable for publication, you may indicate that here to bypass the “Comments to the Author” section, enter your conflict of interest statement in the “Confidential to Editor” section, and submit your "Accept" recommendation.

Reviewer #1: All comments have been addressed

Reviewer #3: (No Response)

Reviewer #4: (No Response)

2. Is the manuscript technically sound, and do the data support the conclusions?

Reviewer #1: Yes

Reviewer #3: Yes

Reviewer #4: Yes

3. Has the statistical analysis been performed appropriately and rigorously? 

Reviewer #1: Yes

Reviewer #3: Yes

Reviewer #4: Yes

4. Have the authors made all data underlying the findings in their manuscript fully available?

Reviewer #1: Yes

Reviewer #3: Yes

Reviewer #4: No

5. Is the manuscript presented in an intelligible fashion and written in standard English?

Reviewer #1: Yes

Reviewer #3: Yes

Reviewer #4: Yes

6. Review Comments to the Author

Reviewer #1: No further edits are needed in the current phase. All comments have been addressed properly by authors.

Reviewer #3: The authors simulated the impacts of future temperature and CO2 conditions as well as predator presence on adult emergence and larval survivorship. They found no impact of the simulated climate conditions or predation risk on larval survivorship, but did find that adult emergence pattern was affected by the simulated climate conditions, with earlier, more uniform emergence under the intermediate climate conditions. This experiment was well-designed and the statistical analyses employed were appropriate. I would make sure that the discussion is clear in which SCCS had the statistically significant differences and why that might be and not conflating the impacts of predator presence/cues from this study and actual predation risk in a non-lab setting.

Reviewer #4: The present paper considers the long-term impacts of climate change on the survival of Aedes aegypti, by the effects on Aedes aegypti itself and the effects of predation on Aedes aegypti, specifically during the larval life stage. The study included simulated climate conditions and testing on Aedes aegypti larvae, with and without the presence of predator risk.

Minor edits

-line 37: define ppmv

-line 63 and throughout: “Zika” is always capitalized

-line 66: change “by” to “during”

Introduction

-line 70: “In addition to… food webs”. This sentence is confusing; isn’t individual biomass an example of a direct effect of climate change on individuals?

-lines 70-90: there is a lot of information in these paragraphs. Some of the information could be moved to the discussion section; overall the Introduction should be more concise. It may help to organize it with some numeric indications, for example “There are three ways in which predator-prey interactions are influenced by climate change….” This may improve the readability of this section.

Methods

-line 140: please include the definition for the abbreviation “ppmv”

-lines 223-233: can you include a brief explanation /example of how skewness/kurtosis represent emergence patterns. Something like: “A left-skewed emergence pattern would indicate the majority of the larvae emerging early on” and “A positive kurtosis would indicate most larvae emerging at the same time”. I think this would help the reader more immediately understand the inclusion of skewness/kurtosis metrics in an analysis of emergence.

Results

-line 250: I think the phrase “emergence pattern” is confusing here – do you mean the day of first emergence? What exactly was being tested here. Generally, it seems that kurtosis and skewness are better representations of the overall emergence pattern. Can you clarify this section?

Discussion

-lines 287-288: do you mean “in some cases, an effect was perceived in the adult stage”

-line 327-329: The result that Intermediate had more of an effect than Extreme SCCS is not too surprising. There is some good literature to show that we would not expect a linear relationship between temperature and life traits. See Mordecai et al. “Thermal biology of mosquito‐borne disease” 2019. Ecology Letters https://doi.org/10.1111/ele.13335

Overall Comments

-the paper is nicely written; there are some areas where the writing could be more concise or more organized to improve readability (see specific comments above)

-another question of interest might be regarding direct predation instead of predation risk; an experiment could test the level of predation between these two species under the different climate scenarios by observing the survival of Aedes aegypti larvae with some predators (with the predators swimming freely outside of a cage). It seems that this would be a better reflection of real-life conditions. Was there a reason to examine predation risk specifically instead of measuring the effect of direct predation in these scenarios?

-The use of skewness and kurtosis to represent the overall emergence pattern is interesting and clever. I think is a great way to summarize emergence patterns. Are there any examples in the literature of these metrics being used to analyze temporal patterns?

7. PLOS authors have the option to publish the peer review history of their article (what does this mean?). If published, this will include your full peer review and any attached files.

Reviewer #1: **Yes: **Mohamed F. Sallam

Reviewer #3: No

Reviewer #4: **Yes: **Rachel Sippy

---

## [Author Response · Author response to Decision Letter 1]

21 Sep 2020

Dear Dr. Jiang-Shiou Hwang,

Academic Editor of PLOS ONE

PONE-D-20-07577R1

Simulated climate change, but not predation risk, accelerates Aedes aegypti emergence in a microcosm experiment in western Amazonia

We are grateful for the valuable suggestions and comments of the reviewers to our manuscript. As requested, we revised the manuscript incorporating the edits suggested by the reviewers. Our point-by-point answers to reviewers comments can be found below. The suggested changes were made directly into the text, and are marked in track changes. 

Reviewer 1:

Comment: No further edits are needed in the current phase. All comments have been addressed properly by authors

Response: Dr. Mohamed F. Sallam, thank you for your previous revision, which greatly improved our manuscript. We are grateful to hear your positive feedback.

Reviewer 3:

Comment: The authors simulated the impacts of future temperature and CO2 conditions as well as predator presence on adult emergence and larval survivorship. They found no impact of the simulated climate conditions or predation risk on larval survivorship, but did find that adult emergence pattern was affected by the simulated climate conditions, with earlier, more uniform emergence under the intermediate climate conditions. This experiment was well-designed and the statistical analyses employed were appropriate. I would make sure that the discussion is clear in which SCCS had the statistically significant differences and why that might be and not conflating the impacts of predator presence/cues from this study and actual predation risk in a non-lab setting.

Response: Thank you for your comment. We answer your comments point-by-point below.

Overall comment 

Abstract

Comment: Line 30: I would remove “those concerning” for clarity.

Response: Changed.

Introduction

Comment: Line 64: I would change to Around the world, some 390 million people are infected with dengue virus each year.

Response: Changed.

Comment: Line 108: Remove mainly

Response: Removed.

Methods

Comment: Line 215-219: This could be made a little clearer if you explicitly state what your response variable is for the linear mixed model. I think it is number of emergences per day, but the reader might not be 100% certain. 

Response: Thank you for your observation. We add “affected the response variable, i.e. number of emergences” (lines 216-217).

Comment: Line 223-234: I think this is a really interesting way to analyze your data and you explained everything very thoroughly.

Response: Thank you for your observation and comment.

Results

Comment: Line 249: Could you specify how much higher the emergence rate was in the Light vs Control SCCS?

Response: We add this information in text, at lines 249-252.

“The first Ae. aegypti adult emergence was on day seven in the Extreme SCCS (n = 3 individuals), on day eight in the Intermediate SCCS (n = 11 individuals), while in the Light and Control SCCS the first adult emerged on day nine (n = 39 individuals; n = 3 individuals, respectively), with higher emergence rate in the Light SCCS.”

Discussion

Comment: Line 279-280: Since you only actually saw the accelerated development time under the intermediate climate change scenario, I would specify that here.

Response: We add this information in the first sentence in Discussion section “We showed that simulated climate change scenarios accelerate development time of Ae. aegypti larvae, which agrees with previous studies in western Amazonia [21], with a emergence peak on a single day in Intermediate SCCS.” (lines 282-284).

Comment: Line 285: Since your larvae weren’t under actual threat of predation (just subjected to chemical cues of predators). I don’t know that you have the grounds to broadly state that climatic variables are a stronger ecological driver than predation risk.

Response: We modified the sentence to make it clear that the effect of SCCS was stronger than predation risk, particularly those related to chemical and visual cues. Now it reads:

“We observed only SCCS did affect adult emergence pattern of Ae. aegypti, indicating that, in this study, climatic variables effects (temperature and CO2 concentration) are stronger ecological driver than predation risk, particularly those related to chemical and visual cues.” (lines 287-290). 

Comment: Line 308: I would substitute predation risk with predation cues here.

Response: Changed. 

Comment: Line 326-327: Should just say “emergence distribution of Ae. aegypti.” without including adults.

Response: “adults” were removed.

Comment: Line 328-329: Since you just saw the effects in the Intermediate treatment, I would be interested to see a discussion of others that have seen a similar non-effect in the Extreme scenario where conditions are too hot and no longer speeding up development.

Response: In the Extreme SCCS, adult emergence started before the other scenarios (see Figure 5). However, when we evaluate the emergence pattern (adult emergence per day for each replicate, captured by kurtosis and skewness) the Intermediate SCCS exhibit a peaked and right-distributed emergence pattern, while this pattern did not occur in the other three SCCS (adult emergence was more evenly distributed throughout the days). This does not mean that extreme conditions did not accelerate the development. We added a sentence in Discussion section including a reference regarding a nonlinear relationship between temperature and life history traits. 

“Typically, the relationship between temperature and life history traits is non-linear [44], and species have a thermal optimum to complete their development. Temperatures above the thermal optimum decrease species performance (e.g. immature survival). Although emergencies started earlier in the Extreme SCCS, the Intermediate SCCS revealed a pronounced emergence on a single day (10th day). (lines 334-338).

Comment: Line 338: Change to “It is important to carry out…”

Response: Changed.

Comment: Line 345: Same as previous comments, you can’t really say Ae. aegypti develops faster under SCCS—just the larvae develop faster under the Intermediate SCCS

Response: Thank you for your observation. We add “larvae” after Ae. aegypti in this sentence, we also reviewed this throughout the text. 

Reviewer 4:

Minor edits

Comment: line 37: define ppmv

Response: We defined ppmv at line 37.

Comment: line 63 and throughout: “Zika” is always capitalized

Response: Thank you for your observation. “Zika” were corrected.

Comment: line 66: change “by” to “during”

Response: Changed.

Introduction

Comment: line 70: “In addition to… food webs”. This sentence is confusing; isn’t individual biomass an example of a direct effect of climate change on individuals?

Response: Thank you for your comment. Yes, the individual biomass is a direct effect of climate change on individuals, we moved this example to the previous line. Now it reads:

 “Climate change affects biodiversity at multiple levels. It may cause shifts on biomass [9], metabolism and behavior [10] at the individual level and, at population level, it can alter species distribution via changes in local conditions. Consequently, community composition can be altered by climate change [11], changing ecosystems and food webs [12–14].” (lines 71-75).

Comment: lines 70-90: there is a lot of information in these paragraphs. Some of the information could be moved to the discussion section; overall the Introduction should be more concise. It may help to organize it with some numeric indications, for example “There are three ways in which predator-prey interactions are influenced by climate change….” This may improve the readability of this section.

Response: Thank you for your example, we used it in the second paragraph of the Introduction section. We also reorganize the ideas in this paragraph to make it more concise and improve the readability, according to your suggestion. Now it reads:

“Climate change affects biodiversity at multiple levels. It may cause shifts on biomass [9], metabolism and behavior [10], at the individual level and, at population level, it can alter species distribution via changes in local conditions. Consequently, community composition can be altered by climate change [11], changing ecosystems and food webs [12–14]. There are two ways in which predator-prey interactions are influenced by climate change. First, it can increase the metabolic rates of individuals, as a consequence of higher temperatures [12], affecting the ability of predators to forage, capture and handle prey. In this way, climate change may modify prey density (density-mediated interactions) [15]; Second, besides direct predation, climate change alter predator-prey interactions via production, transmission, and reception of chemical cues. Under such circumstances, both predator and prey may suffer reduction in their abilities to detect each other [16]. In predation risk situations, releases of chemical cues is common, and the detection of predator by prey through them can modify feeding behavior and/or development rates (trait-mediated interactions) [15].” (lines 71-84).

Methods

Comment: line 140: please include the definition for the abbreviation “ppmv”

Response: The definition of “ppmv” was included.

Comment: lines 223-233: can you include a brief explanation /example of how skewness/kurtosis represent emergence patterns. Something like: “A left-skewed emergence pattern would indicate the majority of the larvae emerging early on” and “A positive kurtosis would indicate most larvae emerging at the same time”. I think this would help the reader more immediately understand the inclusion of skewness/kurtosis metrics in an analysis of emergence.

Response: We add a sentence explained how skewness/kurtosis represent emergence patterns. 

“For example, a right-skewed would indicate an early emergence of Ae. aegypti, whereas a left-skewed would indicate a late emergence.” (lines 225-227)… and “A positive kurtosis would indicate that most larvae emerge at the same time, whereas a negative kurtosis would indicate a more evenly emergence distribution.” (lines 231 -233).

Results

Comment: line 250: I think the phrase “emergence pattern” is confusing here – do you mean the day of first emergence? What exactly was being tested here. Generally, it seems that kurtosis and skewness are better representations of the overall emergence pattern. Can you clarify this section?

Response: We agree that the term “emergence pattern” is confusing here. We replace it to “emergence” only. Here, “emergence” means adult emergence per day for each replicate and not the day of first emergence. We agree that kurtosis and skewness are better representations of the overall emergence pattern, but we used a linear mixed model (LMM) (with Gaussian distribution) to detect if predation and SCCS and their interaction affect number of emergences, controlling for repeated measures and time autocorrelation. In this model we considered the replicates as a random factor (replicates were our repeated measure), with an autocorrelation function to control repetition across days, and SCCS and predation risk as fixed factors. This model is equivalent to a two-way ANOVA with repeated measures. The model output is not informative in terms of temporal patterning, because we did not include time as a fixed factor, so as to retain a viable number of degrees of freedom. To better understand this effect of time without including time in our models as fixed factor, we preferred to use the approach mentioned above (calculate kurtosis and skewness of distribution of emergence time).

Discussion

Comment: lines 287-288: do you mean “in some cases, an effect was perceived in the adult stage”

Response: Yes, you are right. We modify the sentence according to your suggestion, at lines 292-293.

Comment: line 327-329: The result that Intermediate had more of an effect than Extreme SCCS is not too surprising. There is some good literature to show that we would not expect a linear relationship between temperature and life traits. See Mordecai et al. “Thermal biology of mosquito‐borne disease” 2019. Ecology Letters https://doi.org/10.1111/ele.13335

Response: Thank you for your suggestion. We added a sentence in the Discussion section including the idea you gave us and citing this interesting paper. This information is at lines 334-337.

Overall Comments

Comment: the paper is nicely written; there are some areas where the writing could be more concise or more organized to improve readability (see specific comments above)

Response: Thank you for your comment. We reorganized the Introduction section according to your comment regarding lines 70-90. 

Comment: another question of interest might be regarding direct predation instead of predation risk; an experiment could test the level of predation between these two species under the different climate scenarios by observing the survival of Aedes aegypti larvae with some predators (with the predators swimming freely outside of a cage). It seems that this would be a better reflection of real-life conditions. Was there a reason to examine predation risk specifically instead of measuring the effect of direct predation in these scenarios?

Response: Thank you for this comment. Indeed, this is an interesting idea to future experiments. We believe that our experiment is a good reflection of real-life conditions. In a natural environment (lakes, ponds, pools), Ae. aegypti larvae are exposed to both predation and predation risk, and prey behavior and density are affected by direct consumption of predator and by predator presence. Ae. aegypti larvae use different strategies to avoid predators, such as seeking shelter in macrophytes roots. In such habitats, they are under predation risk effects, which can lead prey to decrease the search for food resources and allocate energy in defense instead of development. As one of our focus in this study was to understand whether the predator stimulus (without direct consumption) could affect the Ae. aegypti emergence rates, if we used direct predation, the density of Ae. aegypti larvae could be greatly reduced or even completely consumed by the predator and we would not reach our goal. I

In a meta-analysis study, Preisser et al (2005) assessed the impact of direct predation (density mediated interaction, DMI), and predation risk (trait-mediated interactions, TMIs) on the interaction predator-prey. They showed that intimidation caused by non-lethal predation has high costs for prey, especially in aquatic environments, and is often overlooked.

We do find it important to include this information in the text “Aedes aegypti larvae use different strategies to avoid predators, such as seeking shelter in macrophytes roots. In such habitats, they are under predation risk effects, which can lead prey to decrease the search for food resources and allocate energy in defense instead of development.” (lines 151-155).

Comment: the use of skewness and kurtosis to represent the overall emergence pattern is interesting and clever. I think is a great way to summarize emergence patterns. Are there any examples in the literature of these metrics being used to analyze temporal patterns?

Response: Thank you for your comment. Yes, these metrics were used by other studies to analyze temporal patterns. For example:

- Siqueira et al. 2008. Species richness, abundance, and body size relationships from a neotropical chironomid assemblage: Looking for patterns. Basic and Applied Ecology. doi: https://doi.org/10.1016/j.baae.2007.06.002. 

The authors evaluated temporal variation, using skewness, of body size of a chironomid assemblage.

- Falconi et al. 2020. Effects of data aggregation on time series analysis of seasonal infections. International Journal of Environmental Research and Public Health. doi: https://www.mdpi.com/1660-4601/17/16/5887

The authors used skewness and kurtosis to assess the daily, weekly, and monthly distribution of three respiratory and enteric infections. These data were used as a preliminary analysis to build a model for time series analysis that can be used for various diseases.

---

## [Editor Report · Decision Letter 2]

8 Oct 2020

Simulated climate change, but not predation risk, accelerates Aedes aegypti emergence in a microcosm experiment in western Amazonia

PONE-D-20-07577R2

Dear Dr. Piovezan Borges,

We’re pleased to inform you that your manuscript has been judged scientifically suitable for publication and will be formally accepted for publication once it meets all outstanding technical requirements.

Kind regards,

Jiang-Shiou Hwang, Ph.D.

Academic Editor

PLOS ONE
---

## [Editor Report · Acceptance letter]

12 Oct 2020

PONE-D-20-07577R2 

Simulated climate change, but not predation risk, accelerates *Aedes aegypti* emergence in a microcosm experiment in western Amazonia 

Dear Dr. Piovezan-Borges:

I'm pleased to inform you that your manuscript has been deemed suitable for publication in PLOS ONE. Congratulations! Your manuscript is now with our production department. 

Kind regards, 

on behalf of

Prof. Jiang-Shiou Hwang 

Academic Editor

PLOS ONE